# Elevated Methylglyoxal: An Elusive Risk Factor Responsible for Early-Onset Cardiovascular Diseases in People Living with HIV-1 Infection

**DOI:** 10.3390/v17040547

**Published:** 2025-04-08

**Authors:** Mahendran Ramasamy, Zachary L. Venn, Fadhel A. Alomar, Ali Namvaran, Benson Edagwa, Santhi Gorantla, Keshore R. Bidasee

**Affiliations:** 1Department of Pharmacology and Experimental Neuroscience, University of Nebraska Medical Center, Omaha, NE 68130, USA; mramasamy@unmc.edu (M.R.); zvenn@unmc.edu (Z.L.V.); anamvaran@unmc.edu (A.N.); benson.edagwa@unmc.edu (B.E.); sgorantla@unmc.edu (S.G.); 2Department of Pharmacology, College of Clinical Pharmacy, Imam Abdulrahman Bin Faisal University, Dammam 31441, Saudi Arabia; falomar@iau.edu.sa; 3Environment and Occupational Health, University of Nebraska Medical Center, Omaha, NE 68198, USA; 4Nebraska Redox Biology Center, Lincoln, NE 68503, USA; 5Center for Heart and Vascular Research, Omaha, NE 68198, USA

**Keywords:** HIV-1, cardiovascular diseases, methylglyoxal, glyoxalase-I, aldehyde dehydrogenase, aldo-keto reductase, glutathione, nicotinamide adenine dinucleotide, nicotinamide adenine dinucleotide phosphate, antioxidant transcription factor nuclear factor erythroid 2-related factor 2 (Nrf2)

## Abstract

People living with HIV (PLWH) develop cardiovascular diseases (CVDs) about a decade earlier and at rates 2–3 times higher than the general population. At present, pharmacological strategies to delay the onset of CVDs in PLWH are unavailable, in part because of an incomplete understanding of its molecular causes. We and others recently uncovered elevated levels of the toxic glycolysis and inflammation-induced byproduct methylglyoxal (MG) in plasma from PLWH and from HIV-infected humanized mice (Hu-mice). We also found a reduction in expression of the primary MG-degrading enzyme glyoxalase I (Glo-I) in autopsied cardiac tissues from HIV-1-infected individuals and HIV-1-infected Hu-mice. Increasing the expression of Glo-I in HIV-1-infected Hu-mice not only attenuated heart failure but also reduced endothelial cell damage, increased the density of perfused microvessels, prevented microvascular leakage and micro-ischemia, and blunted the expression of the inflammation-induced protein vascular protein-1 (VAP-1), key mediators of CVDs. In this narrative review, we posit that elevated MG is a contributing cause for the early onset of CVDs in PLWH. Pharmacological strategies to prevent MG accumulation and delay the development of early-onset CVDs in PLWH are also discussed.

## 1. Introduction

In 2014, the United Nations General Assembly’s Declaration on Ending the AIDS epidemic committed countries to the 95–95–95 targets. Two key elements of this declaration were to provide HIV testing and treatment for 90% of people infected with HIV-1 by the end of 2024 [1], and significant progress has been made in achieving these goals globally [2,3,4,5,6,7,8,9]. However, new challenges have emerged, as people with HIV-1 infection on antiretroviral drugs (ARDs) are developing cardiovascular diseases (CVDs) at least a decade earlier than that of the general population and at rates 2–3 times higher [10,11,12,13,14]. Not surprisingly, the prevalence of CVDs is also higher in women living with HIV-1, since female sex is a risk factor for CVDs [9,10,11,12,13,14,15,16,17]. Starting ARDs soon after HIV-1 diagnosis and without interruption [18,19,20]; reducing behavioral/lifestyle risk factors, including smoking, illicit drug use, and alcohol; and treating health risks, including metabolic syndrome, dyslipidemia, hypertension, and renal dysfunction, soon after diagnosis have delayed the onset of CVDs and blunted major adverse cardiovascular events, including acute myocardial infarction, stroke, peripheral artery disease, and sudden cardiac death, in PLWH [18,19,20,21,22,23]. However, why middle-aged HIV-1-infected individuals on ARDs are developing early-onset CVDs remains puzzling.

To date, studies attribute early-onset CVDs in PLWH to “premature/accelerated aging” arising from persistent activation of immune system and inflammation by HIV-1 proteins and dead cell fragments and chronic ARD use [24,25,26]. However, pharmacological strategies to suppress immune activation and inflammation have yielded mixed results, probably because several mechanisms contribute to immune activation and inflammation, and individually targeting each of these inflammatory pathways is not adequate to completely suppress immune activation and inflammation. Moreover, while ARDs efficiently suppress plasma HIV-1 viremia, the tissue distribution of the various classes of ARDs varies, differentially impacting immune activation, inflammation, and oxidative stress in the various tissues/organs. Improvements in tissue bioavailability by altering routes of administration, dosing frequencies, and longer durations of action have also been tested to address these deficiencies [14,27,28,29,30]. These and other data, led others to conclude that the degree of cardiovascular risk in HIV-1 infection cannot be fully explained by persistent immune activation, inflammation, and health- and lifestyle-related risk factors. Unidentified factor(s) are likely playing roles in the development of early-onset CVDs in PLWH [14]. Recently, Ntsekhe and Baker suggested that clonal hematopoiesis of indeterminate potential (CHIP) and the expansion of somatic mutations in leukocytes could worsen the absolute burden of excess CVD risks among PLWH with advancing age [31]. However, since CHIP is age-dependent, it is unlikely to be the underlying cause of early-onset CVDs in PLWH.

We and others recently uncovered elevated levels of the toxic glycolysis and inflammation-induced byproduct methylglyoxal (MG) in plasma from people living with HIV and HIV-infected humanized mice (Hu-mice) [32,33]. We also found decreased levels of the primary MG-degrading enzyme glyoxalase I (Glo-I) in autopsied cardiac tissues from HIV-1 infected individuals and HIV-1 infected Hu-mice. Increasing expression of Glo-I in HIV-1-infected Hu-mice not only blunted heart failure development but also minimized endothelial cell damage, increased the density of perfused microvessels, and prevented microvascular leakage and micro-ischemia [33]. To our surprise, increasing Glo-I expression in HIV-1-infected Hu-mice also decreased expression of inflammation-induced and MG-synthesizing enzyme vascular adhesion protein-1 (VAP-1), suggesting a link between elevated MG, inflammation, and CVDs. In this narrative review, we propose that elevated levels of MG could be an elusive risk factor that is contributing to early-onset CVDs in PLWH. This increase in MG in HIV-1 infection is arising from an increase in its synthesis (via glycolysis and upregulation of VAP-1) and from a decrease in its degradation due to the downregulation of MG-degrading enzymes and co-factors. In this brief review, we also discuss pharmacological strategies to lower MG levels that could blunt early-onset CVDs in PLWH.

## 2. Life Cycle of HIV

HIV-1 primarily infects cells of the immune system that contain the CD4 protein, including CD4^+^-T cells, monocytes, macrophages, dendritic cells, and microglia [34,35,36]. HIV-1 DNA has also been found inside other cells, including epithelial cells (renal and gut), endothelial cells, cardiac myocytes, astrocytes, smooth muscle cells, and pericytes. However, the extent to which the latter cell types support the productive replication of HIV-1 remains unclear [37,38,39,40,41]. HIV-1 also preferentially infects activated CD4^+^-T cells over quiescent CD4^+^-T cells, since activated cells have higher levels of Glut1, higher levels of glycolysis, and increased levels of the HIV-1 co-receptor CCR5 on their plasma membranes [42,43]. For cellular entry, the HIV-1 envelope glycoproteins gp120 and gp41 attach themselves to the CD4 protein, and the co-receptors CCR5/CXCR4 on the plasma membrane of the cell. After entry, HIV-1 capsid uncoats and releases HIV-1 RNA, reverse transcriptase, and HIV-1 integrase into the cytoplasm of the cell. The reverse transcriptase converts the HIV-RNA into HIV-1 DNA, which then enters the nucleus and integrates with the DNA of the infected cell (Figure 1; left side) [44]. HIV subtypes that use CCR5 for cell entry are termed the R5 HIV subtype, those that use CXCR4 are termed the X4 HIV subtype, and those that use both co-receptors are called R5X4 HIV [45,46]. CCR5 is expressed on memory T lymphocytes, macrophages, microglia, dendritic cells, and vascular smooth muscle cells [47], while CXCR4 is found on astrocytes, pericytes, cardiac myocytes, and endothelial cells [48,49]. After integration, HIV-1 genes (Gag and GagPol precursor polyproteins, viral envelope glycoproteins, and regulatory and accessory viral proteins) are transcribed in the Golgi and exported to the cytoplasm and plasma membrane for assembly and release (Figure 1; right side) [44]. Every step of these processes requires energy from the infected cell.

## 3. Energy Substrates

Glucose, glutamine, and fatty acids are the primary substrates utilized by immune cells (CD4^+^T, monocytes, macrophages, dendritic cells, and microglia); vascular cells (endothelial cells, smooth muscle cells, and pericytes); lymphatic cells (lymph endothelial cells and lymphatic smooth muscles); cardiac myocytes; and astrocytes to generate the ATP [50,51,52,53]. Resting immune cells, vascular cells, cardiac myocytes, and astrocytes take up glucose via plasma membrane glucose transporter proteins, including Glut1, and via glycolysis, convert each glucose molecule into two molecules of pyruvate. Two molecules of adenosine triphosphate (ATP) are also generated during the process. In oxygen-rich environments, pyruvate is shuttled to the tricarboxylic acid (TCA) cycle of the mitochondria to generate NADH and FADH2 that fuel oxidative phosphorylation (OXPHOS) in the electron transport chain. In mitochondria, each molecule of pyruvate (in the presence of oxygen) generates 16 molecules of ATP (Figure 2) [51,52]. It should be mentioned that glycolysis produces ATP 100× faster than OXPHOS [53]. Alpha-ketoglutarate (α-KG) is another substrate that is utilized by the TCA cycle for the production of NADH/FADH2 and ATP [52]. α-KG is generated from the breakdown of glutamine, the most abundant circulating amino acid by two sequential enzymatic reactions. Each molecule of α-KG that enters the TCA cycle generates 24 ATP molecules [52]. Beta oxidation of fatty acids also generates and shuttles acetyl CoA to the TCA cycle to generate NADH and FADH2.

Studies have shown that glucose uptake and glycolysis are upregulated in infected CD4^+^-T cells to rapidly provide the substrates needed for HIV-1 integration, replication, assembly, and release [42,54,55]. Others have also shown that glycolysis remains elevated in CD4^+^-T cells isolated from PLWH with low HIV-1 viremia [56]. Following infection, glucose is shunted from glucose-storing organs like skeletal muscles and the liver to accommodate the high glucose demand needed for HIV-1 replication [57]. Naïve CD4^+^-T cells constantly traffic between the blood, peripheral tissues, and lymphatic system in search of foreign antigens. After infection, these cells are responsible for the establishment of HIV-1 reservoirs in lymph nodes [58]. These lymph node reservoirs are usually maintained due to the clonal expansion of latently infected CD4^+^-T cells and limited ARD penetrance [59].

## 4. Methylglyoxal Production

In addition to generating ATP, pyruvate, and other substrates needed for HIV-1 replication, glycolysis also generates the diffusible and highly reactive α-oxoaldehyde methylglyoxal (MG) from the breakdown of the triose intermediates, glyceraldehyde 3-phosphate (G-3-P), and dihydroxyacetone phosphate (DHAP) [60,61]. About 0.1% of all glucose flux is converted into MG [62]. Glycolytic bottlenecks, arising from a reduction in GAPDH and impairment of pathways, that typically shunt glycolytic intermediates, including the hexose monophosphate shunt and polyol pathway, also result in higher levels of G-3-P and DHAP and, by extension, more MG production [63,64]. MG is also synthesized from the breakdown of aminoacetone by the inflammation-induced MG-generating ectoenzyme vascular adhesion protein 1 (VAP-1) [65,66]. The latter is especially important, since VAP-1 is upregulated in vasculature smooth muscle cells (SMCs) during inflammation, resulting in a localized source of MG in proximity to vascular endothelial cells (ECs) (Figure 3) [67,68]. Earlier, we reported that ECs have half as much of Glo-I per μg of cells compared to SMCs [68]. Smaller amounts of MG are also generated from fatty acid oxidation; however, the extent to which this occurs in HIV-1 infection with and without ART remains poorly characterized.

## 5. Degradation of Methylglyoxal

In eucaryotes, MG is degraded via three mechanisms: the glyoxalase system (glyoxalase I, Glo-I, and glyoxalase I, Glo-II), with co-factor-reduced glutathione (GSH) and water [69]; aldehyde dehydrogenases (ALDHs; oxidation reaction), with nicotinamide adenine dinucleotide as a co-factor; and aldo-keto reductases, with GSH and/or nicotinamide adenine dinucleotide phosphate as co-factors [69,70]. The expression of Glo1, ALDHs, and AKR and the enzymes that synthesize their co-factors are regulated by the antioxidant transcription factor nuclear factor (erythroid-derived 2)-like 2 (Nrf2)–antioxidant response element (ARE) [71,72,73,74]. It should be mentioned that Glo-I/Glo-II are more widely distributed than ALDHs and AKRs, and the kinetics of degradation of MG by Glo-I/Glo-II are faster. Studies have also linked the downregulation/knockout of these enzymes to impairment in cardiovascular functions [75,76,77].

### 5.1. MG Degradation via Glyoxalase System

The primary pathway for the degradation of MG is the two-enzyme glyoxalase system. In the first step, the rate-limiting Glo-I converts a hemithioacetal formed between MG and reduced glutathione (MG-GSH) into S,D-lactoylglutathione [69]. In the second step, Glo-II, in the presence of water, degrades S,D-lactoylglutathione into D-lactic acid and GSH (Figure 4). Giacco et al. showed that the knockdown of Glo-I mimics diabetic nephropathy in nondiabetic mice [77]. Dobariya et al. showed that the deletion of Glo-I exacerbates acetaminophen-induced hepatotoxicity in mice [78]. The knockdown of Glo-I in mouse aortic endothelial cells resulted in impairment of angiogenesis [79]. Glo1 knockout reduces anxiety-like behavior but increases depression-like behavior in mice [75]. Others have shown that overexpression of Glo-I reduces hyperglycemia-induced levels of advanced glycation end products and oxidative stress and improves neurovascular coupling and endothelial dysfunction in diabetic rats [80,81]. Others have shown that Glo-I knockout in zebrafish results in upregulated ALDH activity [82]. In yeast, the loss of Glo1 results in hypersensitivity to MG and death [83].

Earlier, we reported downregulation of Glo1 in the hearts of humanized mice infected with HIV-1 (NOD.Cg-Prkdc(scid)Il2rg(tm1Wjl)/SzJ), in plasma from PLWH, and in autopsied cardiac tissues from deceased HIV seropositive individuals [33]. These findings should not come as a surprise, since several groups have reported the downregulation of Nrf2 in HIV-1 infection [84,85,86,87]. For example, Fan et al. showed that HIV-1-related proteins downregulate Nrf2 expression and/or activity within the alveolar epithelium, which in turn impairs antioxidant defenses and barrier function, rendering the lungs susceptible to oxidative stress and injury [84]. Han et al. also found that Nrf2 activation blocks HIV replication in macrophages before the integration phase and inhibits the expression of apoptotic pathways in virus-exposed macrophages [85]. Staitieh et al. also showed that HIV-1 infection and exposure to HIV-1-related proteins inhibit Nrf2–ARE activity in alveolar macrophages and impair their phagocytic function [86]. Davinelli et al. also found a significant reduction in the protein levels of Nrf2 and hemooxygenase-1 (HO-1) in HIV-1 transgenic rats, suggesting a weakening in the protection exerted by the Nrf2/HO-1 system [87].

### 5.2. MG Degradation via Oxidation

ALDHs are another class of enzymes that is responsible for the NAD(P)-dependent oxidation of aldehydes, including MG, to carboxylic acids [69]. The E1, E2, and E3 isozymes of ALDHs oxidize MG into pyruvate in a NAD-dependent manner (Figure 5). Murine models with a loss of ALDHs have reported elevated aldehydic adducts, cardiovascular and motor dysfunction, and tissue damage. Single-nucleotide polymorphisms in ALDH2 and mutations in ALDH1/2 are also linked to increases in risk of CVDs [76,88]. Conversely, overexpression of ALDHs attenuates the effects of oxidative stress and reactive oxygen species (ROS) in various organs, conditions that are often upregulated in response to MG accumulation [89,90]. The knockout of the Glo1 protein in zebrafish also results in ALDH activity. However, MG levels remain moderately elevated in Glo1 knockout zebrafish, suggesting that ALDHs and AKRs are not sufficient replacements for Glo1 [82]. Vander Jagt and Hunsaker indicated that ALDHs and AKRs have lower affinities for MG compared to Glo1 [70]. To date, the effects of HIV-1 on the expression of ALDHs remain poorly characterized.

### 5.3. MG Degradation via Reduction

AKRs are a group of enzymes that reduce aldehydes and ketones into their corresponding alcohols [91,92]. AKR-mediated MG degradation operates through two pathways: a GSH-dependent pathway, where the hemithioacetal formed between GSH and MG is converted by an AKR and NADPH to lactaldehyde, and a GSH-independent pathway, where MG reacts with the AKR and NADPH to form acetol [92,93]. Further metabolism of lactaldehyde and acetol by the AKR leads to the production of propanediol (Figure 6). In the presence of GSH, the efficiency of reduction in MG by ALDHs increases but the site of reduction switches from the aldehyde to the ketone carbonyl. The latter arises because glutathione converts ALDHs from an aldehyde reductase to a ketone reductase [94].

AKRs can also increase MG levels by increasing the production of the triose phosphate intermediate dihydroxyacetone phosphate [93,94]. Aldose reductase is the first enzyme of the polyol pathway, and it reduces glucose to sorbitol, which can then be converted to fructose by sorbitol dehydrogenase. Fructose is then converted to fructose-1-phosphate and, eventually, to DHAP and MG (Figure 7). To date, the effects of HIV-1 on the expression of AKRs remain poorly characterized.

## 6. Consequences of Elevated MG

Under non-oxidative/non-inflammatory environments, the majority of MG generated in cells is reversibly bound to the SH moieties on GSH and cysteine residues of proteins [95]. The hemithioacetal formed between MG and GSH (MG-GSH) is also targeted for degradation by Glo-I. Under oxidative/inflammatory conditions, as is the case in HIV-1 infection, where Nrf2 expression is downregulated and steady state levels of Glo-I and GSH are reduced, MG will accumulate. This MG will irreversibly react with accessible basic moieties on proteins, DNA/RNA, and lipids. On proteins, MG reacts with arginine, lysine, and histidine residues. On arginine residues, the primary adduct formed is the hydroimidazolone MG-H1 (>90% of all MG adducts). Lesser amounts of MG-H2, MG-H3, [96] and *N*^δ^-(5-hydroxy-4,6-dimethylpyramidine-2-yl)-l-ornithine argpyrimidine (AP) have also been detected [97] (Figure 8). When MG reacts to lysine residues, *N*^ε^-(1-carboxyethyl) lysine (CEL) and the lysine dimer 1,3-di(*N*^ε^-lysino)-4-methyl-imidazolium (MOLD) are formed [98]. These adducts have been identified on proteins in the cytoplasm, and inside the Golgi, mitochondria, lysosomes, and endoplasmic reticulum, as MG, can diffuse from the cytoplasm into these organelles. NH moieties on certain lipids, including ceramide and sphingosine, may also be chemically modified by MG. MG will also diffuse from the cytoplasm into the nucleus to form adducts with basic moieties on DNA. The N2 position of the deoxyguanosine (dG) nucleotide is most susceptible to MG modification, leading to the formation of dihydroimidazolone, *N^2^*-(1,2-dihydroxy-2-methyl) ethano-dG (cMG-dG) and *N*^2^-(1-carboxyethyl)-7-1-hydroxy-2-oxopropyl-dG (MG-CEdG) [99]. MG-CEdG is a good marker for long-term MG accumulation. No enzymes have been identified in eukaryote cells that are capable of breaking MG adducts after they are formed.

### The Link Between Elevated MG and CVDs in HIV-1 Infection

Heart failure, essential hypertension, pulmonary hypertension, and stroke are common CVDs in PLWH [100,101,102,103]. Studies suggest that these diseases are due, in part, to the dysregulation of vasculature ECs and the build up of plaques inside arteries arising from immune activation, inflammation, oxidative stress, and certain classes of ARDs. To date, clinical trials to suppress immune activation and inflammation to suppress CVDs have yielded mixed results, in part because of an incomplete understanding of their molecular triggers [104,105,106,107,108,109,110,111,112]. As a result, pharmacological strategies to protect the endothelium and reduce arterosclerosis in PLWH remain limited.

We and others showed earlier that exposing vascular ECs to MG reduces the expression of tight junction proteins and induced apoptosis. Bathing cerebral pial arterioles of anesthetized rats with MG decreased their response to an EC-mediated vasodilator [68]. Exposing ECs to MG also decreases the phosphorylation of eNOS at Ser^1177^, causing eNOS to switch from the generation of nitric oxide (NO) to superoxide anion [113]. The dysregulation of vascular ECs will lead to microvascular leakage, a reduction in the density of perfused microvessels, and micro-ischemia [68,114,115,116]. Tissue micro-ischemia will induce the expression of hypoxia-inducible factor 1-alpha (HIF-1α) to upregulate glycolysis genes, MG production, and tissue fibrosis. Others have shown that MG will (i) induce lipid oxidation and accumulation in macrophages (foam cell formation) [117]; (ii) trigger the proliferation of vascular smooth muscle cells (vSMCs), decreasing blood vessel diameter [118]; (iii) induce intima thickening and increase arterial wall stiffness [119,120]; and (iv) induce platelet hyperaggregation and reduce thrombus stability [121]. Thus, elevated MG could be contributing to atherosclerosis plaque buildup and hypertension in PLWH.

MG is an established activator of multiple inflammation pathways, including NOD-like receptor protein inflammasomes, nuclear factor kappa-light-chain-enhancer of activated B cells (NF-κB), activator protein-1 (AP-1), and the receptor for advanced glycation end products (RAGE) [101,122,123,124,125,126,127,128,129]. These data could help explain why mono-pharmacology to attenuate inflammatory mediators, including interleukin-1β (IL-1β) and tumor necrotic factor-alpha (TNF-α), have yielded mixed results. Our group also showed that MG adducts can impair cardiac contraction and relaxation by forming adducts on the type 2 ryanodine receptor calcium release channel (RyR2), sarco(endo)plasmic reticulum Ca^2+^-ATPase (SERCA 2), and myosin heavy chains (MHC—alpha and beta) [130,131,132]. These adducts dose-dependently increase and then reduce the activity of RyR2, triggering the depletion of Ca^2+^ from the sarcoplasmic reticulum (SR), delaying the return of Ca^2+^ into the SR, and decreasing the rate and force of myocyte contraction. Others have shown that MG can impair the function of other ion channels, including Na^+^ and K^+^ channels. We also showed that elevated MG was responsible, in part, for exacerbated ischemia–reperfusion injury in the brains of rats following mid-cerebral artery occlusion [68]. Although the specific reason for this was not clear, we suspect that MG increases ROS generation in mitochondria via complex I and complex III. Studies have linked elevated MG to kidney disease, certain cancers, retinopathy, and sepsis [20,130,133,134,135,136,137,138,139,140,141,142]. Recent data suggest that elevated MG may also contribute to the high incidence of pulmonary arterial hypertension (PAH) and right ventricular dysfunction in PLWH by increasing arterial stiffening via the cross-linking of collagen within the extracellular matrix (ECM) [143].

## 7. Therapeutic Strategies to Lower Methylglyoxal Levels

Recently, we and others uncovered elevated MG in the plasma of PLWH, autopsied tissues from HIV-infected individuals, and tissues and plasma from HIV-1-infected Hu-mice [32,33]. We also found an increase in the expression of the inflammation-induced and MG-synthesizing enzyme VAP-1 and a decrease in the expression of Glo-I in autopsied cardiac tissues from HIV-1-infected individuals and HIV-1-infected Hu-mice. We also showed that increasing the expression of Glo-I in HIV-1-infected Hu-mice using a custom-engineered adeno-associated virus designed to express Glo-I under inflammatory/oxidative conditions not only blunted heart failure, but it also attenuated cardiac endothelial damage, increased the density of perfused microvessels, attenuated microvascular leakage and micro-ischemia, and decreased the expression of VAP-1. The latter is consistent with a reduction in inflammation [33]. Taken collectively, these data suggest that lower MG could attenuate early-onset CVDs in PLWH.

Several strategies have been tested in preclinical models and clinical trials to lower MG, attenuate inflammation, and blunt CVDs in other diseases. These include treatment with agents that contain nucleophilic moieties (arginine/arginine analogs) to which MG can preferentially react and prevent the formation of adducts on proteins, lipids and DNA; treatment with agents to prevent or reduce the formation of MG by diverting early glycolysis substrates to the polyol and other pathways; treatment with agents that can increase the expression of Nrf2 and MG-degrading enzymes; and treatment with agents that increase the levels of the co-factors needed by MG-degrading enzymes, including reduced glutathione, nicotinamide adenine dinucleotide, and nicotinamide adenine dinucleotide phosphate, including polyphenols and flavonoids/isoflavonoids (Figure 9 and Table 1). Recently, we also replaced the endogenous promoter of AAV2/9 with the promoter of preproendothelin1 to be able to drive the expression of Glo-I under inflammatory conditions [33,68,144].

### 7.1. Treatment with Agents Containing Nucleophilic Groups to Scavenge MG

MG is an electrophile and, when elevated, chemically reacts with amino moieties on proteins, lipids, and DNA/RNA to form adducts. Agents that contain amino moieties were tested in animal models to determine their ability to scavenge MG. Agents tested in diabetes models include aminoguanidine, pyridoxamine, L-carnosinol, and arginine-containing linear and cyclic peptides. Aminoguanidine was shown to lower AGE formation and prevent diabetic nephropathy retinopathy, neuropathy, and heart failure in preclinical models but was not as less effective in clinical trials [166]. Pyridoxamine, a naturally occurring B vitamin, was also found to attenuate diabetic nephropathy and preserves renal function [145,146] in rodent models of diabetes and patients with Type 2 diabetes. It also decreased the formation of AGEs, inflammation, and pain in patients with osteoarthritis. The naturally occurring histidine dipeptide l-carnosine also lowers MG levels in rodent models of metabolic syndrome and normalized insulin sensitivity, insulin secretion, and glucose tolerance in overweight and obese subjects [147,167,168,169]. However, its therapeutic potential is limited due to a rapid hydrolysis of the peptide by carnosinases. Carnosinol, a derivative of L-carnosine that is resistant to metabolism by carnosinase [147,148], dose-dependently reduced carbonyl stress, normalized glycemic control, and mitigated inflammation and steatohepatitis [147]. The 40-amino acid arginine-containing peptide (GERP_10_) lowered MG. However, GERP_10_ was rapidly degraded by endogenous peptidases. To improve the half-life, the cyclic arginine-containing peptide CycK(Myr)R4E was developed [149]. CycK(Myr)R4E was found to have comparable MG scavenging activity as GERP_10_ and prevented MG-induced adducts and pain in mice [149]. The widely used antidiabetic agent metformin contains a guanidine, and its ability to scavenge MG could help explain, in part, its ability to reduce vascular complications in animal and clinical studies [170]. A major limitation of MG-scavenging molecules is that their use must be continuous, as they are unlikely to get rid of the underlying inflammation and oxidative stress. Because high does of these agents are needed, additional benefits could be obtained from non-MG scavenging mechanisms. Inadequate biodistribution/bioavailability could limit the usefulness of MG-scavenging approaches.

### 7.2. Treatment with Agents to Prevent or Reduce the Formation of MG

Benfotiamine is an activator of transketolase, which directs glucose-6-phosphate to the pentose phosphate pathway (hexose monophosphate shunt). Studies have shown that treatment of diabetic rodents with benfotiamine decreased plasma MG and MG-derived glycation and microvascular complications [152]. In individuals with Type 2 diabetes, treatment with a high dose of thiamine (the parent compound) both prevented and reversed early-stage diabetic nephropathy [171]. Benfotiamine also attenuated NF-κB activation [150]. Since activated NF-κB suppresses Nrf2 mRNA expression, reducing NF-κB activation could enhance Nrf2 levels and the expression of antioxidant enzymes. Wu and colleagues showed earlier that the upregulation of aldolase B, a key enzyme in fructose metabolism, increased vascular MG production [172]. They also showed that insulin-enhanced MG overproduction in insulin-sensitive adipocytes was due to upregulating aldolase A. Thus, it is possible that the inhibition of aldolase A and aldolase B could suppress MG production and CVDs. Caloric restriction and exercise training have also been shown to reduce MG and improve vascular function in individuals with Type 2 diabetes and diabetic rats [153,154,155]. The MG-lowering effect of caloric restriction and exercise was attributed to better metabolic control. Studies have also shown that metformin suppresses MG production by reducing hepatic gluconeogenesis.

VAP-1 is an inflammation-induced enzyme that catalyzes the deamination of aminoacetone to produce MG, hydrogen peroxide, and ammonia. Several VAP-1 inhibitors have been evaluated in preclinical and clinical studies, including MDL72974A, PXS-4728A, SZE5302, and LJP1586. These studies demonstrated that the inhibition of VAP-1 attenuated inflammation, intracerebral hemorrhagic stroke in mice, neurologic dysfunction in rats subjected to subarachnoid hemorrhage, atherosclerosis, diabetic macular edema, nephropathy and retinopathy, and ischemic reperfusion injury [156,157,158]. In diabetic rat models, the inhibition of VAP-1 attenuated the development of atherosclerotic lesions independent of serum glucose levels. There are also ongoing clinical studies evaluating the effects of the VAP-1 inhibitor on reducing inflammation associated with non-alcoholic steatohepatitis (TT-01025-CL) [159]; primary sclerosing cholangitis (BTT1023, a monoclonal antibody that blocks VAP-1) [173]; and chronic kidney disease in patients with Type 2 diabetes (ASP8232) [157]. However, to the best of our knowledge, MG levels or MG adducts in SMCs have not been measured in VAP-1 inhibition studies. Since the expression of VAP-1 is triggered by inflammation and many inflammatory pathways are activated in PLWH, including AP-1, NF-κB, NRLP inflammasomes, and the receptor for advanced glycation end products (RAGE), it is not clear whether VAP-1 inhibitors would be sufficient for lowering inflammation/oxidative stress in PLWH.

### 7.3. Treatment with Agents to Increase Expression of Nrf2 and MG-Degrading Enzymes

Nrf2 is the principal transcription factor that drives the expression of the antioxidant and MG-degrading enzymes Glo1, ALDHs, and AKRs and their co-factors. In healthy cells, Nrf2 is continuously synthesized and released into the cytoplasm. Keap1 is a thiol-rich E3 ubiquitin ligase that tightly regulates the activity of Nrf2 by binding to and ubiquitinating it for proteasome-dependent degradation. Under oxidative stress, the cysteine residues on Keap1 become oxidized, allowing Nrf2 to escape ubiquitination and translocate to the nucleus, and induces the expression of antioxidant genes.

Sulfur-containing compounds, polyphenols, and flavonoids/isoflavanoids, including sulforaphane (broccoli, Brussels sprouts, and kale); curcumin (turmeric); resveratrol (grapes, red wine, and berries); epigallocatechin gallate (green tea); and quercetin (apples, onions, and berries) (Figure 9) have been identified as “Nrf2 inducers” [160,161,162,174,175]. In cells, these nucleophiles are converted into electrophiles by metabolic enzymes, including polyphenol oxidases (PPOs) [176]. The “electrophiles” then covalently react with the cysteine residues in Keap1, causing it to dissociate from Nrf2 [177]. Nrf2 then translocates to the nucleus where it binds to the antioxidant response element (ARE) of DNA to trigger the expression of antioxidant genes that protect against oxidative stress. The perseverant tert-butylhydroquinone and the FDA-approved drugs dimethyl fumarate, used to treat sclerosis and psoriasis; bardoxolone methyl, used to treat chronic kidney disease and certain cancers; and carbamazepine, used to treat epilepsy and bipolar disorders, also dissociate Keap1 from Nrf2 [178,179]. A drawback of “Nrf2 enhancers” is that under inflammatory conditions, as is the case in HIV-1, activated NF-κB will suppress Nrf2 mRNA translation, resulting in a reduction in Nrf2 expression [180,181]. Thus, it is likely that the beneficial effects of Nrf2 enhancers will be significantly reduced over time in chronic inflammatory conditions. Another drawback could be the limited bioavailability and tissue distribution of polyphenols and flavonoids/isoflavanoids

Overexpressing ALDHs and AKRs has also been shown to attenuate MG levels [151,182]. Thornalley and colleagues also showed that treating overweight and obese subjects with *trans*-resveratrol and hesperetin (tRES-HESP) to induce the expression of Glo-I also attenuated MG. They also found that while the tRES-HESP combination reversed insulin resistance, this improvement was not seen in individuals treated with with tRES and HESP individually, suggesting that pharmacological synergism between tRES-HESP and that HESP was likely inhibiting intestinal glucuronosyl transferase [183]. Recently, we replaced the endogenous promoter of AAV2/9 with the promoter of preproendothelin1 and show that we can induce the expression of Glo-I under inflammatory conditions. Using this strategy, we were not only able to blunt heart failure in HIV-1-infected Hu-mice and a rat model of Type 1 diabetic rats but also attenuated cardiac endothelial damage, increased the density of perfused microvessels, attenuated microvascular leakage and micro-ischemia, and decreased the expression of VAP-1 [33,68,144].

### 7.4. Treatment to Increase Co-Factors for MG-Degrading Enzymes

Oxidative stress is elevated in PLWH due to impairment in the antioxidant defense system and from the downregulation of Nrf2 [184,185]. The downregulation of Nrf2 will not only decrease the expression of antioxidant enzymes but also decreases the levels of co-factors needed by MG-degrading enzymes, including reduced glutathione, nicotinamide adenine dinucleotide, and nicotinamide adenine dinucleotide phosphate, and enzymes involved in their synthesis, including a in reduction γ-glutamate cysteine ligase glutathione synthetase and NAD(P)H:quinone oxidoreductase 1 (NQO1) and nicotinamide N-methyltransferase and NAD kinase [186]. Several over-the-counter (OTC) antioxidant supplements are available either as a single agent or as a combination with antioxidants, including N-acetylcysteine (a precursor to GSH), vitamin C, vitamin E, lipoic acid, polyphenols, flavonoids/isoflavanoids, green tea extract, Coenzyme Q10, curcumin (turmeric), pomegranate, etc. Vegetable and fruit extracts that contain compounds that activate Nrf2 (see Section 7.3 above) are also available OTC. Selective FDA-approved drugs to lower CVD risks, such as statins and angiotensin receptor blockers, also have antioxidant properties [163,164]. Kumar et al. [165] found that PLWH treated with a combination of glycine and N-acetyl cysteine (GlyNAC) for 12 weeks improved several outcomes, including red-blood cell and muscle GSH concentrations, mitochondrial function, mitophagy and autophagy, oxidative stress, inflammation, endothelial function, genomic damage, insulin resistance, glucose production, muscle protein breakdown rates, body composition, physical function, and cognition. Juruga et al. [187] also found that supplementation of vitamins A, C, and E in PLWH decreased modified DNA bases and partially restored antioxidant enzymes (superoxide dismutase and catalase). In another study, Silva et al. [188] found serum triglyceride levels were increased after curcumin supplementation in PLWH. Erdos et al. [189] indicated that available data are not sufficient to conclude whether supplementation with glutathione would attenuate HIV-associated neurocognitive impairment in PLWH. Wilkinson et al. [190] also concluded that antioxidant micronutrients, alone or in combination, in PLWH have yielded mostly null results, with a few studies observing mild benefits of supplementation with zinc [191,192,193] and selenium [194,195,196]. The poor absorption of antioxidants from the gut, low levels of other micronutrients (selenium and zinc), and a low dose of the antioxidant may be reasons for variations in efficacies. It is also possible that the reduction in endogenously generated antioxidants is the consequence of other changes that are occurring in PLWH and that supplementation with exogenous antioxidants is not sufficient to alleviate the antioxidant deficit.

## 8. Conclusions

Highly active antiretroviral drugs have transformed HIV-1 infection from a literal death sentence to a chronic manageable disease. However, new challenges have emerged as PLWH are developing an array of early-onset CVDs for which the underlying causes remain poorly understood. Here, we posit that an elevation in glycolysis and the inflammation-induced byproduct MG could be an elusive risk factor that is triggering early-onset CVDs in PLWH. The increase in MG is triggering (i) a dysregulation of vascular endothelial cells and pericytes, resulting in microvascular leakage, a reduction in the density of perfused microvessels, micro-ischemia, and fibrosis; (ii) a proliferation of vascular smooth muscle cells that could decrease the blood vessel diameter, leading to hypertension; (iii) intima thickening, leading to an increase in arterial wall stiffness; (iv) lipid oxidation and accumulation in macrophages (foam cell formation), leading to vascular plaques; (v) platelet hyper-aggregation and reduced thrombus stability; (vi) impairment in cellular ionic homeostasis; and (vii) an enhancement in oxidative stress and inflammation. We also theorize that elevated MG could impair the function of lymphatic endothelial cells and lymphatic smooth muscles and by extension impair the function of the lymphatic vasculature. Although clinically approved interventions for MG-associated complications are not currently available, some treatments are available as nutritional supplements which could be used to lower MG levels and reduce CVD risks in PLWH.

## Figures and Tables

**Figure 1 viruses-17-00547-f001:**
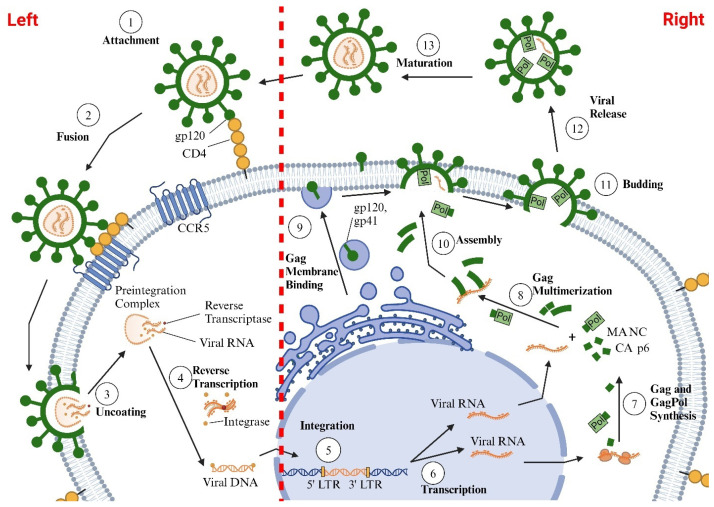
Illustration of the HIV-1 life cycle, which is divided into two main phases: an early phase and a late phase. The early phase consists of several sequential steps: (i) HIV initially attaches to the cell via its envelope glycoproteins, gp120 and gp41, which bind to the CD4 receptor on the cell surface; (ii) gp120 and gp41 then engage with chemokine receptors, CXCR4 and CCR5, facilitating fusion with the cell membrane and viral entry; (iii) once inside, the HIV-1 capsid uncoats in the cytoplasm, releasing HIV-1 RNA, reverse transcriptase (to convert HIV RNA into HIV DNA), and integrase (to integrate HIV DNA into the host cell’s DNA) (**left side**). The late phase includes the steps following the integration of HIV-1 DNA into the host DNA: (i) transcription of HIV genes; (ii) export of HIV-1 RNAs from the nucleus to the cytoplasm and translation of these RNAs to produce Gag and GagPol precursor polyproteins, envelope glycoproteins, and regulatory and accessory proteins; (iii) transport of Gag, GagPol, and envelope glycoproteins to the plasma membrane; (iv) assembly of the Gag and GagPol polyproteins on the host cell’s plasma membrane; (v) encapsidation of the viral RNA genome by the forming Gag lattice; (vi) incorporation of viral Env glycoproteins; and (vii) budding of new virions from the host cell, followed by particle maturation (**right side**).

**Figure 2 viruses-17-00547-f002:**
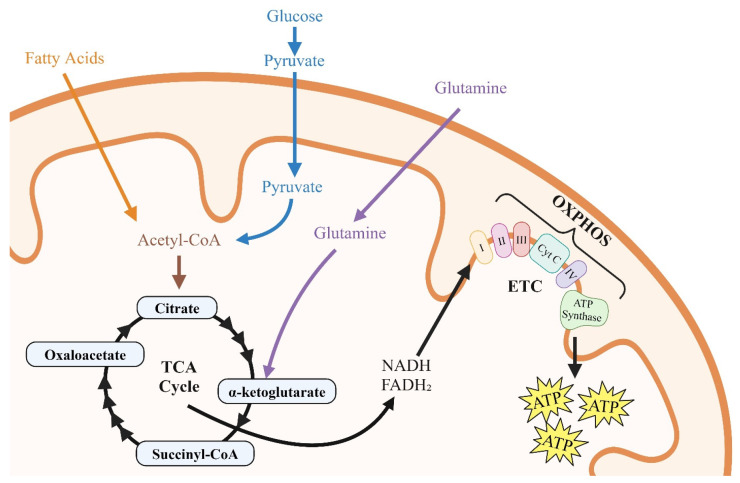
Illustration of the roles of glycolysis, fatty acids, and glutamine in cellular metabolism. In CD4^+^-T cells, glucose uptake initiates glycolysis, producing two molecules of ATP and two molecules of pyruvate. The pyruvate is then transported into the mitochondria, entering the tricarboxylic acid (TCA) cycle, where it generates NADH and FADH_2_. These molecules drive oxidative phosphorylation (OXPHOS) and the electron transport chain (ETC), resulting in the production of up to 16 molecules of ATP per molecule of pyruvate.

**Figure 3 viruses-17-00547-f003:**
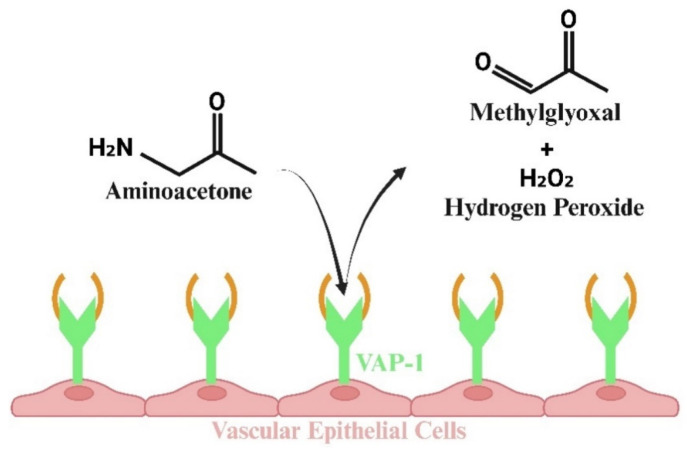
Illustration showing that HIV replication results in the synthesis of methylglyoxal (MG) via the glycolysis pathway. Additionally, MG is produced by the inflammation-induced ectoenzyme vascular adhesion protein 1 (VAP-1) through the breakdown of aminoacetone. This process is particularly important, as VAP-1 is upregulated in the vasculature, particularly in vascular smooth muscle cells, leading to elevated localized concentrations of MG near vascular endothelial cells.

**Figure 4 viruses-17-00547-f004:**
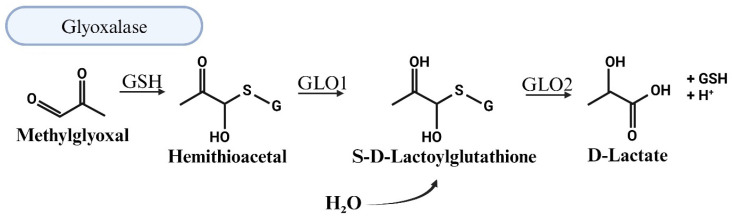
Illustration of the detoxification of methylglyoxal (MG) through the glyoxalase system. The main pathway for methylglyoxal (MG) degradation involves the two-enzyme glyoxalase system. In the initial step, the rate-limiting enzyme glyoxalase-1 (Glo1) catalyzes the conversion of a hemithioacetal formed between MG and reduced glutathione (MG-GSH) into S,D-lactoylglutathione. Subsequently, the second enzyme, glyoxalase-2 (Glo2), acts in the presence of water to degrade S,D-lactoylglutathione into D-lactic acid and glutathione (GSH).

**Figure 5 viruses-17-00547-f005:**
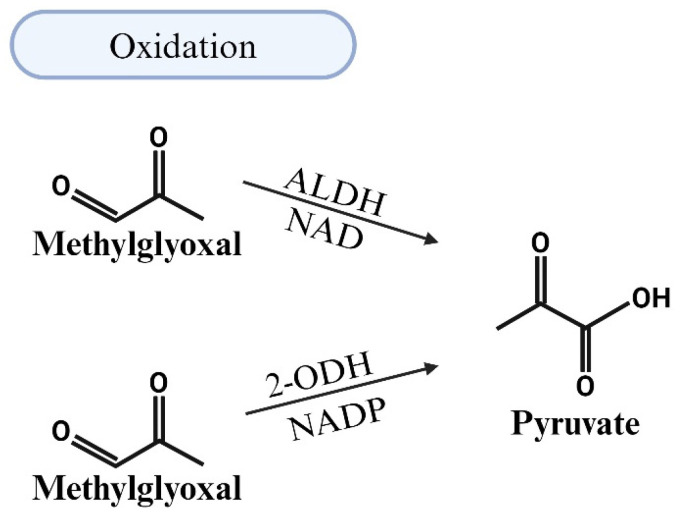
Illustration of the detoxification of methylglyoxal (MG) through oxidation by aldehyde dehydrogenases (ALDHs). This class of enzymes is responsible for the NAD(P)-dependent oxidation of aldehydes, including MG, into carboxylic acids. ALDHs facilitate the oxidation of MG to pyruvate in a NAD-dependent manner.

**Figure 6 viruses-17-00547-f006:**
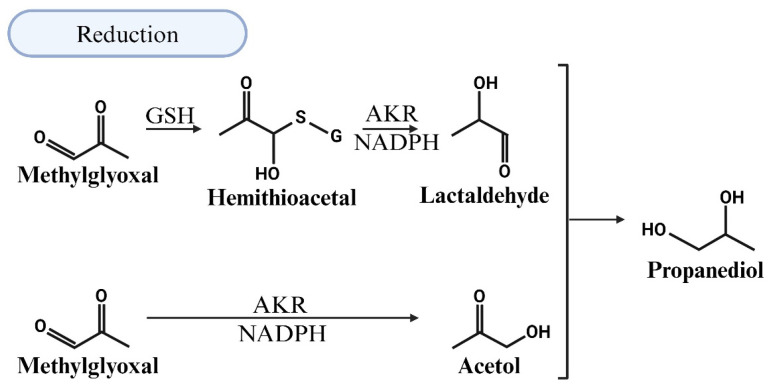
Illustration of the detoxification of methylglyoxal (MG) through reduction in the presence of reduced glutathione (GSH). The efficiency of MG reduction by aldehyde dehydrogenases (ALDHs) increases, but the reduction site shifts from the aldehyde to the ketone carbonyl. This shift occurs because glutathione modifies ALDHs, converting them from aldehyde reductases to ketone reductases. Further metabolism of the resulting lactaldehyde and acetol by aldo-keto reductase (AKR) leads to the production of propanediol.

**Figure 7 viruses-17-00547-f007:**
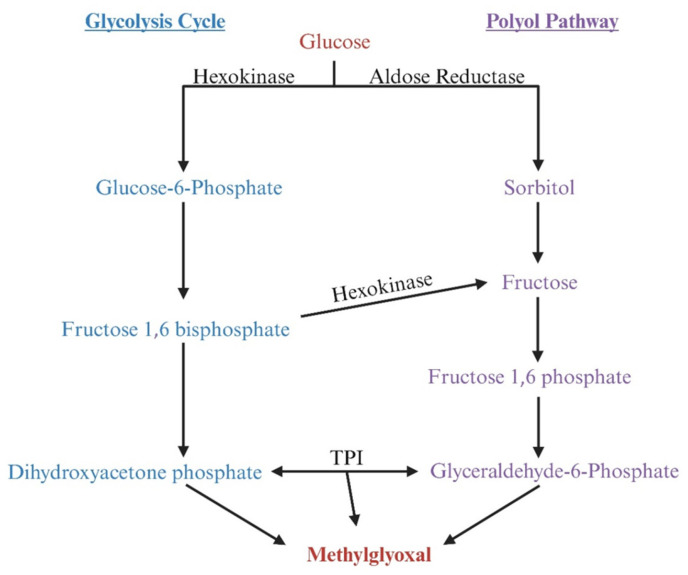
Illustration demonstrating that aldo-keto reductase (AKR) can degrade methylglyoxal (MG) but also indirectly promotes MG production by increasing the production of the triose phosphate intermediate dihydroxyacetone phosphate (DHAP). Aldose reductase (AR), the first enzyme in the polyol pathway, reduces glucose to sorbitol, which is subsequently converted to fructose by sorbitol dehydrogenase. Fructose is then phosphorylated to fructose-1-phosphate by ketohexokinase, which is further converted to DHAP by fructose bisphosphate aldolase. Alternatively, fructose can be phosphorylated to fructose-6-phosphate by hexokinase, which is then converted to fructose-1,6-bisphosphate by phosphofructokinase-1; this compound is subsequently converted to DHAP by fructose bisphosphate aldolase. DHAP can yield MG.

**Figure 8 viruses-17-00547-f008:**
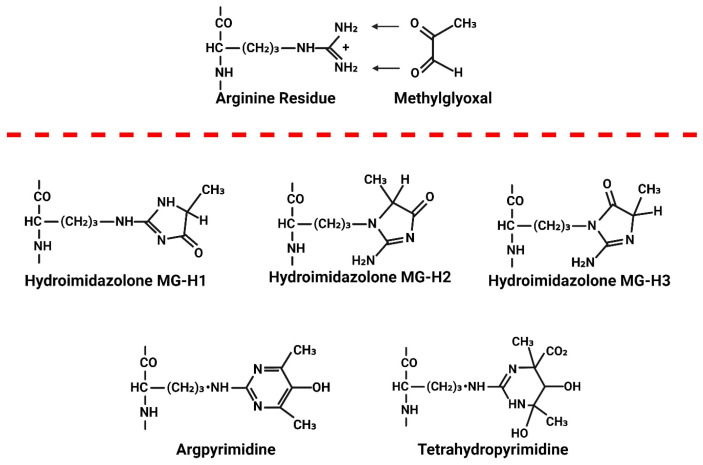
Illustration showing the formation of methylglyoxal (MG) adducts on proteins. When MG levels are elevated, it irreversibly reacts with accessible arginine, lysine, and histidine residues on proteins. This reaction with arginine results in the formation of three hydroimidazolone adducts: MG-H1, MG-H2, and MG-H3. Methylglyoxal (MG) also reacts with arginine to form *N*^δ^-(5-hydroxy-4,6-dimethylpyrimidine-2-yl)-L-ornithine, known as argpyrimidine (AP), and *N*^δ^-(4-carboxy-4,6-dimethyl-5,6-dihydroxy-1,4,5,6-tetrahydropyrimidine-2-yl)-L-ornithine, known as tetrahydropyrimidine (THP).

**Figure 9 viruses-17-00547-f009:**
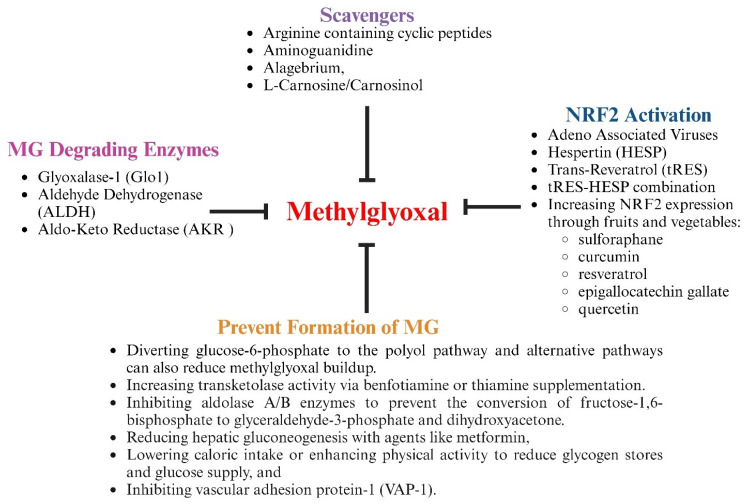
Illustration showing pharmacological strategies to blunt MG accumulation.

**Table 1 viruses-17-00547-t001:** Therapeutic strategies to lower methylglyoxal levels.

Strategy	Agents and Mechanism of Action	Advantages	Drawbacks	References
Agents containing nucleophilic groups to scavenge MG	Aminoguanidine, pyridoxamine, L-carnosinol, and arginine-containing peptides to scavenge MG by forming adducts with nucleophilic moieties.	(1) Effective in reducing scavenging MG; (2) Can also improve insulin sensitivity and glucose tolerance and reduces inflammation.	(1) Limited success in clinical trials, probably due to their inability to adequately lower oxidative stress and inflammation; (2) Some agents, like carnosinol, are degraded by endogenous enzymes; limiting long-term efficacy; (3) Inadequate biodistribution and bioavailablity, limiting usefulness.	[145,146,147,148,149]
Agents to prevent or reduce MG formation	(1) Benfotiamine (activates transketolase); (2) Metformin (lowers gluconeogenesis); (3) Caloric restriction and exercise.	(1) Effective in preclinical models and some clinical studies; (2) Improves metabolic control and vascular function; (3) Benfotiamine also attenuates NF-κB activation, reducing inflammation.	(1) Variability in clinical outcomes; dependency on metabolic state; (2) Caloric restriction and exercise require sustained adherence.	[150,151,152,153,154,155]
VAP-1 inhibitors	(1) MDL72974A, PXS-4728A, and LJP1586, which reduce MG production by inhibiting VAP-1.	(1) Demonstrated anti-inflammatory effects; (2) Potential to reduce vascular complications; reduces oxidative stress and prevents endothelial dysfunction.	(1) Limited data on direct MG measurement in these studies; (2) Ongoing clinical trials; (3) Potential for off-target effects; (4) May not sufficiently lower inflammation and oxidative stress.	[156,157,158,159]
Nrf2 activators and MG-degrading enzyme upregulation	(1) Sulforaphane, curcumin, resveratrol, and quercetin, which act as Nrf2 enhancers; (2) MG-degrading enzymes (Glo-I, ALDHs, and AKRs).	(1) Shown to lower MG levels in preclinical models; (2) Abundant in natural foods; (3) Protective against oxidative stress and inflammation; (4) Ability to insert an inflammation gene promoter to drive expression of antioxidant genes under oxidative and inflammatory conditions. (5) Promoters of inflammation-induced proteins can be used to drive expression of Glo-I, ALDHs and AKRs using adeno-associated vectors (6) Single IV injection can last for 6 month	(1) Small molecules, “Nrf2 enhancers”, may have limited bioavailability and tissue distribution; (2) Nrf2 expression may be downregulated under inflammatory conditions; some polyphenols require dietary adjustments for effectiveness. (3) Clinical studies would be needed to test efficacy and safety of adeno-associated viruses (AAV) in humans (4) Efficacy of AAVs may be reduced over time	[33,68,144,160,161,162]
Increasing co-factors for MG-degrading enzymes	Supplementation with antioxidants like glutathione and vitamins C and E and co-factors such as NAD(P)H.	(1) Shown to improve mitochondrial function, reduce oxidative stress, and enhance endothelial function in some PLWH studies; (2) GlyNAC supplementation improved cognition, muscle strength, and oxidative stress markers.	(1) Inconsistent clinical outcomes; (2) Poor absorption of some antioxidants; potential need for combination therapies; (3) Some studies showed no improvement in MG levels despite antioxidant supplementation.	[163,164,165]

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
