# Peer review of "Elevated Methylglyoxal: An Elusive Risk Factor Responsible for Early-Onset Cardiovascular Diseases in People Living with HIV-1 Infection"

_viruses, 2025, doi:10.3390/v17040547_

Round 1

Reviewer 1 Report

Comments and Suggestions for Authors

A brief summary)

With the advancement of ART, the life expectancy of PLWH has increased, making the prevention and management of comorbidities increasingly important. It is also crucial to prevent and manage cardiovascular disease from this perspective. PLWHs are at a higher risk of early onset of these diseases compared to immunocompetent individuals, making it important to identify risk factors and prevent disease progression. In this regard, this study might be noteworthy.

General concept comments)

The introduction is somewhat lengthy. The section discussing that early initiation of ART can prevent cardiovascular disease could be condensed. Instead, it would be beneficial to elaborate more on the background explaining why the study focuses on increased MG levels as the risk factor of CVD in PLWH.

Additionally, it would be helpful to clarify whether these MG-related findings are a cause or a consequence of early-onset cardiovascular disease in PLWH.

Also, discussing whether these findings also contribute to cardiovascular disease in immunocompetent individuals would enhance the manuscript's broader relevance.

Specific comments)

In the beginning of the Introduction section, the manuscript refers to the UNAIDS targets, but the most recent update in 2024 has revised the goal to 95-95-95. It would be beneficial to update this information accordingly to ensure accuracy and alignment with the latest global standards.

Author Response

Reviewer # 1 

Comment #1: With the advancement of ART, the life expectancy of PLWH has increased, making the prevention and management of comorbidities increasingly important. It is also crucial to prevent and manage cardiovascular disease from this perspective. PLWHs are at a higher risk of early onset of these diseases compared to immunocompetent individuals, making it important to identify risk factors and prevent disease progression. In this regard, this study might be noteworthy.

Response: Thank you

Comment#2: The introduction is somewhat lengthy. The section discussing that early initiation of ART can prevent cardiovascular disease could be condensed. Instead, it would be beneficial to elaborate more on the background explaining why the study focuses on increased MG levels as the risk factor of CVD in PLWH.

Response: As per the suggestion, have focused the introduction on MG levels and added in a section where we link elevated MG with CVDs.

Comment#3: It would be helpful to clarify whether these MG-related findings are a cause or a consequence of early-onset cardiovascular disease in PLWH.

Response: Earlier, we showed that administering a custom-generated adeno-associated virus designed to express the MG-degrading enzyme glyoxalase I (Glo-I) under inflammatory/oxidative conditions, blunted cardiac dysfunction in HIV-infected humanized mice with suggesting that elevated MG is an contributing cause  of early-onset cardiovascular disease (Front Cardiovasc Med. 2021 Dec 14:8:792180).  It should be pointed out that  these HIV-1 infected humanized mice were not treated with antiretroviral drugs.

Comment #4: Discussing whether these findings also contribute to cardiovascular disease in immunocompetent individuals would enhance the manuscript's broader relevance.

Response: We appreciate this comment. Although oxidative stress and inflammation can be elevated in immunocompetent individuals, (Medicina Intensiva Volume 38, Issue 2, March 2014, Pages 73-82), and glycolysis is elevated in immunocompetent cells, it is not clear if glycolysis is also enhanced in immunocompetent individuals. For MG to be elevated, both synthesis (glycolysis) and degradation (downregulation of Glo-I) must be present. 

Comment #5: In the beginning of the Introduction section, the manuscript refers to the UNAIDS targets, but the most recent update in 2024 has revised the goal to 95-95-95. It would be beneficial to update this information accordingly to ensure accuracy and alignment with the latest global standards.

Response: As per your suggestion, we have revised the goal to be 95-95-95 (with the new citation).  Thank you

Reviewer 2 Report

Comments and Suggestions for Authors

The current review by Ramasamy et al describes the potential elevated Methylglyoxal as a risk factor responsible for Early-Onset Cardiovascular Diseases in People Living With HIV-1 Infection. The review is comprehensive, well written, and addresses important current cardiac issues in people living with HIV-1 Infection.  The review can be enhanced with following minor points.

  1. Include a summary table in section 7 with description of each strategy proposed to down regulate the Methylglyoxal by mentioning advantages and drawbacks.

  1. Please describe if there are epidemiology studies with statical significance correlating elevated level of Methylglyoxal in HIV infected plasma available in the literature. Can plasma level Methylglyoxal can be used as a marker for screening for as early-onset cardiovascular diseases in people living with HIV-1 infection? How reliable would that be with the complexity of many factors involved in the onset of cardiac issues in HIV infected individuals?

Author Response

Reviewer #2

Comment #1: The current review by Ramasamy et al describes the potential elevated Methylglyoxal as a risk factor responsible for Early-Onset Cardiovascular Diseases in People Living With HIV-1 Infection. The review is comprehensive, well written, and addresses important current cardiac issues in people living with HIV-1 Infection.  The review can be enhanced with following minor points.

Response:  Thank you.  We have made the requested corrections

Comment #2: Include a summary Table with description of each strategy proposed to down regulate the Methylglyoxal by mentioning advantages and drawbacks.

Response: We have incorporated a summary Table in Section 7, detailing the proposed strategies to downregulate methylglyoxal levels, titled "Therapeutic Strategies to Lower Methylglyoxal Levels".  We have also summarized advantages and weaknesses of each of the approaches.

Comment #2: Please describe if there are epidemiology studies with statistical significance correlating elevated level of methylglyoxal in HIV infected plasma available in the literature. Can plasma level Methylglyoxal can be used as a marker for screening for as early-onset cardiovascular diseases in people living with HIV-1 infection? How reliable would that be with the complexity of many factors involved in the onset of cardiac issues in HIV infected individuals?

Response:  In August 2021, Kamari et al., published a reported showing elevated methylglyoxal-derived hydroimidazolone 1 (MG-H1) adduct in serum in PLWH on antiretroviral drugs (tenofovir disoproxil fumarate/emtricitabine plus atazanavir/ritonavir, darunavir/ritonavir, or raltegravir over 96 weeks. In December of 2021, we also reported elevated levels of MG-H1 adducts in plasma of PLWH.  In that study we also showed that administering an AAV designed to overexpress the MG-degrading enzyme glyoxalase-1 under inflammatory/oxidative stress conditions in HIV-1 infected humanized mice attenuated heart failure (Front Cardiovasc Med. 2021 Dec 14:8:792180).  While these are the only two articles today showing elevated MG in PLWH, there are several review articles implicating elevated MG in cardiometabolic disorders, vascular complications, blood brain barrier integrity, some of which are cited in the text.

  1. Allaman, I.; Belanger, M.; Magistretti, P. J., Methylglyoxal, the dark side of glycolysis. Front Neurosci 2015, 9, 23.
  2. Alomar, F. A., Methylglyoxal in COVID-19-induced hyperglycemia and new-onset diabetes. Eur Rev Med Pharmacol Sci 2022, 26, (21), 8152-8171.
  3. Angeloni, C.; Zambonin, L.; Hrelia, S., Role of methylglyoxal in Alzheimer's disease. Biomed Res Int 2014, 2014, 238485.
  4. Beisswenger, P. J., Methylglyoxal in diabetes: link to treatment, glycaemic control and biomarkers of complications. Biochem Soc Trans 2014, 42, (2), 450-6.
  5. Bellahcene, A.; Nokin, M. J.; Castronovo, V.; Schalkwijk, C., Methylglyoxal-derived stress: An emerging biological factor involved in the onset and progression of cancer. Semin Cancer Biol 2018, 49, 64-74.
  6. Berdowska, I.; Matusiewicz, M.; Fecka, I., Methylglyoxal in Cardiometabolic Disorders: Routes Leading to Pathology Counterbalanced by Treatment Strategies. Molecules 2023, 28, (23).
  7. Berends, E.; van Oostenbrugge, R. J.; Foulquier, S.; Schalkwijk, C. G., Methylglyoxal, a highly reactive dicarbonyl compound, as a threat for blood brain barrier integrity. Fluids Barriers CNS 2023, 20, (1), 75.
  8. Bourajjaj, M.; Stehouwer, C. D.; van Hinsbergh, V. W.; Schalkwijk, C. G., Role of methylglyoxal adducts in the development of vascular complications in diabetes mellitus. Biochem Soc Trans 2003, 31, (Pt 6), 1400-2.
  9. Chang, T.; Wu, L., Methylglyoxal, oxidative stress, and hypertension. Can J Physiol Pharmacol 2006, 84, (12), 1229-38.
  10. Sankaralingam, S.; Ibrahim, A.; Rahman, M. D. M.; Eid, A. H.; Munusamy, S., Role of Methylglyoxal in Diabetic Cardiovascular and Kidney Diseases: Insights from Basic Science for Application into Clinical Practice. Curr Pharm Des 2018, 24, (26), 3072-3083.

 Thank you